# Specific patterns of vital sign fluctuations predict infection and enable sepsis diagnosis in pediatric burn patients

Farzin Sadeq[1,2,3,4], Jonah Poster[1,3,4,5], Chris Chu[1,2,3,4], Joan Weber[1,2,4], Martha Lydon[1,2,4], Maggie Dylewski Begis[1,2,4], Robert Leo Sheridan[1,2,4]*, Korkut Uygun[1,2,3,4]*

1 Department of Research, Shriners Hospitals for Children Boston, Boston, Massachusetts, United States of America, 2 Department of Surgery, Massachusetts General Hospital, Boston, Massachusetts, United States of America, 3 Center for Engineering in Medicine and Surgery, Massachusetts General Hospital, Boston, Massachusetts, United States of America, 4 Harvard Medical School, Boston, Massachusetts, United States of America, 5 Icahn School of Medicine at Mount Sinai, New York City, New York, United States of America

* kuygun@mgh.harvard.edu (KU); rsheridan@mgh.harvard.edu (RLS)

**Data Availability Statement:** Data cannot be shared publicly because of Protected Health Information (PHI) identifiers. Western Institutional Review Board (WIRB) has imposed sharing

## Abstract

Early recognition of the clinical signs of bloodstream infection in pediatric burn patients is key to improving survival rates in the burn unit. The objective of this study was to propose a simple scoring criteria that used readily available temperature, heart rate (HR) and mean arterial pressure (MAP) data to accurately predict bloodstream infection in pediatric burn patients. A retrospective chart review included 100 patients admitted to the pediatric burn unit for >20% total body surface area (TBSA) burn injuries. Each patient had multiple blood culture tests, and each test was treated as a separate and independent "infection event" for analysis. The time at each blood culture draw was time 0 for that event, and temperature, HR and MAP data was collected for 24 hours after the blood culture was drawn. "Infection events" included in this study had at least six complete sets of temperature, HR and MAP data entries. Median temperature, HR and MAP, as well as mean fever spikes, HR spikes and MAP dips, were compared between infection group (positive blood cultures) and control group (negative blood cultures). These vital sign fluctuations were evaluated individually and as a combination of all three as timely predictors of bloodstream infection. In addition, we tested the prediction of Gram-negative bacteria versus Gram-positive or fungi present in blood cultures. Patients in the infection group had significantly higher median temperatures ($p<0.001$), mean fever spikes ($p<0.001$) and mean HR spikes ($p<0.001$), compared to the control group. Using the combination scoring criteria to predict bloodstream infection, the strongest predictive values in the 24-hour timeframe had high sensitivity (93%) and specificity (81%). The predictive test metric based on vital sign spikes predicted Gram-negative bacteria, but with limited sensitivity (57%) and specificity (44%). A simple scoring criteria using a combination of fever spikes, HR spikes and MAP dips predicted bloodstream infection in pediatric burn patients, and can be feasibly implemented in routine clinical care. There is also potential to use the predictive metric to detect a few select organisms based on vital signs, however further work is necessary to enhance accuracy to levels that would allow consideration for clinical use.

restrictions on the data set containing Protected Health Information (PHI) identifiers. Further queries regarding data access can be referred to CRSTsubmissions@shrinenet.org.

**Funding:** The authors KU and RLS received funding from the Shriners Hospitals for Children - Boston Clinical Research Core, Shriners Hospitals for Children (Clinical Research Grant 71004), which is gratefully acknowledged. The funders had no role in study design, data collection and analysis, decision to publish, or preparation of the manuscript.

**Competing interests:** The authors have declared that no competing interests exist.

## Introduction

Sepsis is among the leading causes of morbidity and mortality in pediatric burn patients [1–5]. Rates of burn-specific sepsis mortality have been reported to be as high as 55% in pediatric populations [6]. While there have been great advances in burn care, diagnosing sepsis in severely burned patients remains a challenge [7]. Patients with large total body surface area (TBSA) burn injuries often manifest systemic inflammatory response syndrome (SIRS), which significantly overlaps with the clinical signs of sepsis [6–9]. Delays in sepsis diagnoses and timely antibiotic administration may result in septic shock, respiratory failure, multiple organ dysfunction syndrome and ultimately death [8–11].

Currently, there are no accurate validated diagnostic tests predicting bloodstream infection in pediatric burn patients [6–11]. The clinical identification of bloodstream infection is limited to standard sepsis criteria and positive blood culture documentation [6–11]. Previous studies have evaluated clinical biomarkers, such as procalcitonin, C-reactive protein, tumor necrosis factor-$\alpha$, interleukin-6, and interleukin-10, to predict bloodstream infection [12–17] and to distinguish Gram-negative bacteria from Gram-positive bacteria and fungi in positive blood cultures [15]. However, these inflammatory markers are typically present in SIRS, even in the absence of bloodstream infection [18–20].

In a previous study, our group proposed a simple scoring criteria using temperature data, where we identified strong predictive accuracy for fever spikes above 39˚C in pediatric burn patients with positive blood cultures [21]. The objective of this study was to improve upon the previous scoring criteria by implementing readily available temperature, heart rate (HR) and mean arterial pressure (MAP) clinical data. By utilizing these dynamic trends in vital signs, there is an opportunity to predict bloodstream infection in pediatric burn patients.

## Methods

### Study design

The current study was approved by Western Institutional Review Board (WIRB) under 45 CFR 164.512. The retrospective chart review of 100 pediatric patients included patients from our previous study that were admitted to the burn unit between 2010 and 2014 for TBSA burn injuries greater than 20%. Each patient had multiple blood cultures drawn during their stay in the pediatric burn unit, and each test was treated as a separate and independent "infection event" for analysis. The time at each blood culture draw was noted as time 0 for that event, and temperature, HR and MAP data was collected for 24 hours after the blood culture was drawn. "Infection events" included in this study had at least six complete sets of temperature, HR and MAP entries collected during the 0–24 hour study timeframe post-blood culture draw. The final sample included 95 pediatric patients with 204 blood cultures. Five patients were excluded from the sample because they did not have at least six complete sets of temperature, HR and MAP entries collected during the 24-hour timeframe post-blood culture draw. The study sample was then divided into two groups based on their documented laboratory blood culture results: infection group included positive blood culture results and control group included negative blood culture results.

Patient demographics and burn injury characteristics were captured from the medical record. Demographic and burn characteristics collected included age, gender, mean percent TBSA and mean percent full thickness burn. Outcome data collected included length of stay (LOS), days on mechanical ventilation, and days in the intensive care unit (ICU). Vital signs data collected included body temperature, HR and MAP. Infection variables collected included

number of positive and negative bloodstream infection cases, names and characteristics of infectious microorganisms present in the positive blood cultures.

All clinical data and laboratory blood cultures were ordered by clinical staff as a part of the standard care treatment at our institution. Blood cultures were drawn largely from central venous catheters. Temperature reading methods included oral, axillary, tympanic and rectal, with rectal temperature measurements being the most frequently used method in our pediatric burn unit. HR reading methods included apical, monitor, and peripheral pulse. MAP reading methods included arterial line, cuff, and bedside medical device interface.

## Statistical analysis

All statistical analyses were performed using Microsoft Excel (Redmond, WA). The Student's *t*-test compared mean age, mean percent TBSA, mean percent full thickness burn, mean LOS, mean mechanical ventilation days, and mean ICU days between the infection and control groups. Median temperature, HR and MAP were compared between the infection and control groups within the 0–24 hour timeframe post-blood culture draw, as well as the mean number of fever spikes, HR spikes and MAP dips. The differences in vital sign measurements between the infection and control groups were evaluated using the *F*-test to determine equal or unequal variances and the Student's *t*-test to determine statistical significance. The Sidak method was also used to correct for multiple comparisons, which controls for the Type 1 error rate. Multivariate regression analyses were performed to determine the weight of each independent variable relative to the other variables. Statistical significance was defined as a *p*-value less than 0.05.

Predictive accuracy of bloodstream infection was tested using the presence of 2 or more fever spikes above 39˚C, 2 or more HR spikes above two standard deviation age-specific norms and 1 or more MAP dips below 60 mmHg. If a patient case satisfied the criteria, bloodstream infection was predicted as positive. Predictive accuracy of Gram-negative bacteria was tested using the presence of 2 or more fever spikes above 39˚C, 14 or more HR spikes above two standard deviation age-specific norms and 5 or more MAP dips below 60 mmHg. If a blood culture satisfied the criteria, Gram-negative bacteria was predicted as positive. The following predictive metrics were then evaluated: true positive rate (sensitivity), true negative rate (specificity), positive predictive value (precision), negative predictive value, false positive rate, false negative rate, false discovery rate, accuracy, and Matthews correlation coefficient (MCC). The MCC determined the degree of agreement (range from -1 to 1) between our predictive metrics and the observed data.

## Results

The infection group consisted of 49 patients with 103 blood cultures with one or more positive bloodstream infections. The control group consisted of 46 patients with 101 blood cultures. The mean ± standard deviation age of patients in the infection group was 7.6±5.8 years with a mean percent TBSA of 48.1±18.2. The mean age of patients in the control group was 6.0±4.4 years with a mean percent TBSA of 23.9±17.1. The majority of the infection group (55%) and the control group (54%) were male. There were statistically significant differences between the groups in percent TBSA (*p*<0.001), percent full thickness burn (*p*<0.001), LOS (*p*<0.001), ventilation days (*p*<0.001) and ICU days (*p*<0.001; Table 1). Four patients from the infection group died in the hospital. All four patients died outside of the 0–24 hour timeframe.

Median temperatures, HR and MAP were compared between the two groups to test whether each median vital sign could predict the presence of bloodstream infection (Table 2). The infection group had statistically significantly higher median temperatures (*p*<0.001)

**Table 1. Patient demographics, burn characteristics and hospital course outcomes.**

| | Patients (N) | Cases (N) | Blood Culture | % Male | Age in years, Mean (SD) | % TBSA, Mean (SD) | % Full Thickness Burn, Mean (SD) | LOS in days, Mean (SD) | Ventilation Days, Mean (SD) | ICU Days, Mean (SD) |
|---|---|---|---|---|---|---|---|---|---|---|
| **Infection** | 49 | 103 | Positive | 55 | 7.6 ± 5.8 | 48.1 ± 18.2 | 43.1 ± 18.2 | 57.7 ± 36.9 | 8.0 ± 10.1 | 32.7 ± 20.1 |
| **Control** | 46 | 101 | Negative | 54 | 6.0 ± 4.4 | 23.9 ± 17.1 | 18.5 ± 15.8 | 34.1 ± 19.2 | 2.2 ± 4.2 | 15.3 ± 10.0 |
| **P-value** | | | | | 1.4E-01 | 3.1E-09[a] | 9.1E-10[a] | 1.8E-04[a] | 4.4E-04[a] | 8.4E-07[a] |

[a]Sidak-corrected statistical significance using Student's t-test

compared to the control group in the 0–24 hour timeframe. However, the difference in median temperature was only 0.5˚C between the two groups. The infection group had a higher but non-significant median HR compared to the control group. Patients in the infection group also had a lower but non-significant median MAP compared to the control group. The temperature, HR and MAP charts of two representative patient cases matched by age are presented in Fig 1.

## Fever spikes

The mean number of fever spikes above 39˚C was compared between the infection and control groups as a predictive measure for bloodstream infection (Table 3). Fever spikes were defined as a temperature above 39˚C [7]. Patients in the infection group experienced significantly more fever spikes within the 0–24 hour timeframe post-blood culture draw, compared to the control group ($p<0.001$). Out of the 204 cases, 35 infection cases and 58 control cases did not have a temperature above 39˚C. Out of the 204 cases, 17 infection cases and 31 control cases had a hypothermic event, which was defined by temperatures below 36.5˚C [7]. The representative patient cases from the infection group (Fig 1A) and control group (Fig 1B) both presented 4 fever spikes above the 39˚C threshold in the 0–24 hour timeframe.

## Heart rate spikes

The mean number of HR spikes was also compared between the infection and control groups as a predictive measure of bloodstream infection (Table 3). HR spikes were defined as the number of beats per minute (BPM) above two standard deviation age-specific norms [7]. The number of HR spikes within 24 hours post-blood culture draw in the infection group is significantly greater compared to the control group ($p<0.001$). Out of the 204 cases, 2 infection cases and 12 control cases did not have HR spikes above two standard deviation age-specific norms. The representative patient case from the infection group (Fig 1C) and the control group (Fig 1D) both presented 13 HR spikes above the 130 BPM age-specific threshold in the 0–24 hour timeframe.

**Table 2. Median temperature, heart rate and mean arterial pressure.**

| | | | | Temperature | | | Heart Rate | | | Mean Arterial Pressure | | |
|---|---|---|---|---|---|---|---|---|---|---|---|---|
| | Patients (N) | Cases (N) | Blood Culture | Median (˚C) | F-test | P-value | Median (BPM) | F-test | P-value | Median (mmHg) | F-test | P-value |
| **Infection** | 49 | 103 | Positive | 38.2 | 4E-02 | 3E-09[a] | 140.0 | 9.7E-01 | 8E-02 | 71.7 | 4E-01 | 7E-01 |
| **Control** | 46 | 101 | Negative | 37.7 | | | 135.7 | | | 72.2 | | |

[a]Sidak-corrected statistical significance using Student's t-test

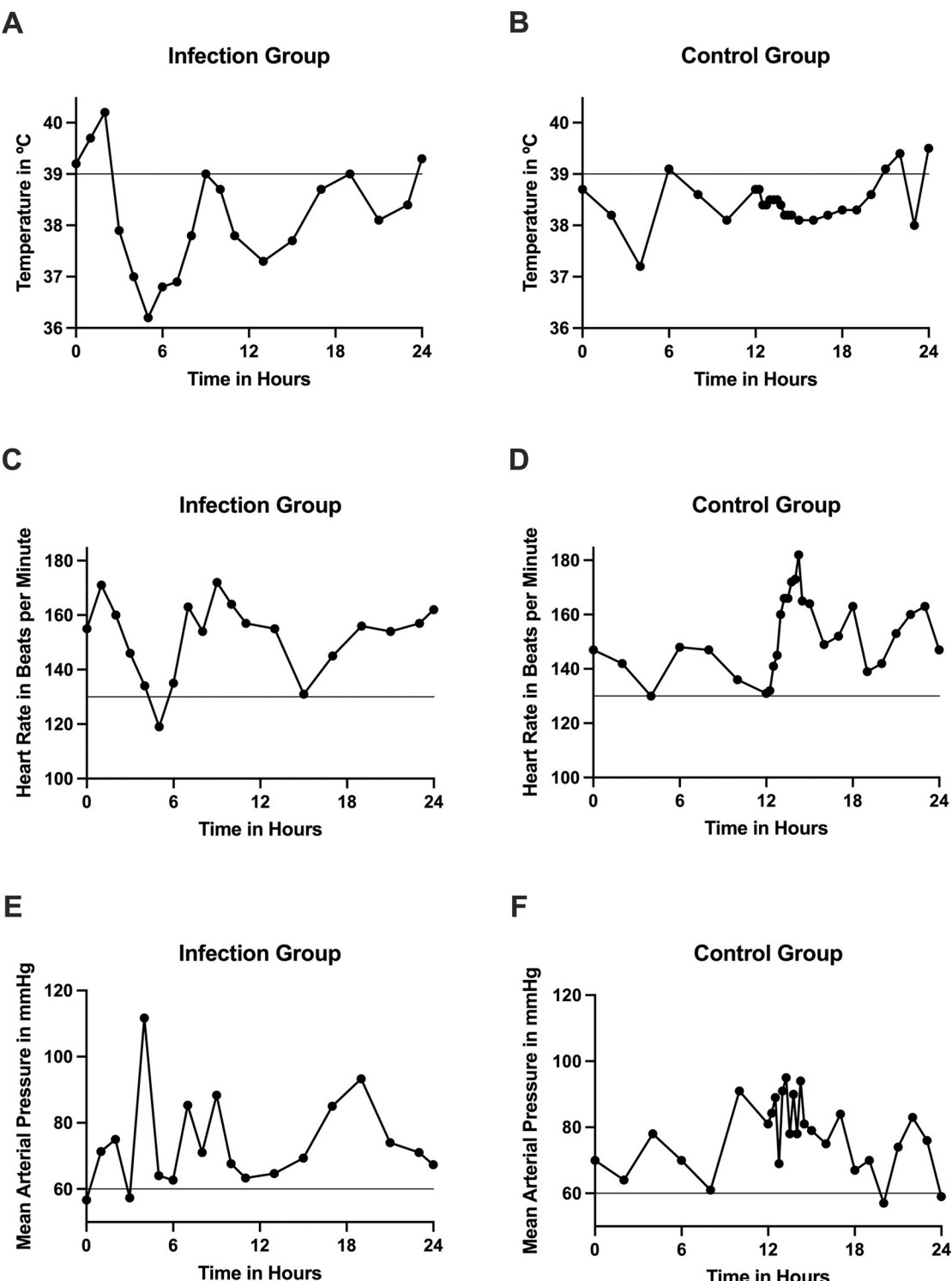

**Fig 1. Temperature, HR and MAP trendlines of representative patients from infection and control groups matched by age.** (A) Temperature trendline of infection group patient. (B) Temperature trendline of control group patient. (C) Heart rate trendline of infection group patient. (D) Heart rate trendline of control group patient. (E) Mean arterial pressure trendline of infection group patient. (F) Mean arterial pressure trendline of control group patient.

**Table 3. Mean fever spikes, heart rate spikes, mean arterial pressure dips.**

| | Patients (N) | Cases (N) | Blood Culture | Temperature | | | Heart Rate | | | Mean Arterial Pressure | | |
|---|---|---|---|---|---|---|---|---|---|---|---|---|
| | | | | Mean Fever Spike (SD) | F-test | P-value | Mean HR Spike (SD) | F-test | P-value | Mean MAP Dip (SD) | F-test | P-value |
| **Infection** | 49 | 103 | Positive | 2.87 ± 3.83 | 5E-14 | 2E-05[a] | 12.2 ± 8.82 | 1E-05 | 5E-08[a] | 4.30 ± 6.18 | 3E-02 | 2E-01 |
| **Control** | 46 | 101 | Negative | 1.03 ± 1.74 | | | 6.33 ± 5.65 | | | 3.25 ± 4.98 | | |

[a]Sidak-corrected statistical significance using Student's t-test

## Mean arterial pressure dips

The mean number of MAP dips was evaluated between the infection and control group as a predictive measure of bloodstream infection (Table 3). MAP dips were defined as MAP below 60 mmHg [7]. The number of MAP dips within 24 hours post-blood culture draw was not statistically significant between the infection and control groups. Out of the 204 cases, 42 infection cases and 40 control cases did not have MAP dips. The representative patient cases from the infection group (Fig 1E) and control group (Fig 1F) both presented 2 MAP dips below the 60 mmHg threshold in the 0–24 hour timeframe.

## Predictive analysis using vital sign patterns

Specific patterns of fever spikes, HR spikes, and MAP dips appeared to have an easily detectable difference within the first 24 hours post-blood culture draw. We performed two multivariate regression analyses for the purpose of analyzing which vital sign variables were more critical for the predictive metric. Our regression analyses revealed that HR spikes were significantly correlated to bloodstream infection and held more statistical weight in the predictive metric than MAP dips or fever spikes. We evaluated each vital sign individually, as well as a combination of all three as a predictor of bloodstream infection. The combination of 1 or more MAP dips and 2 or more fever spikes or 2 or more HR spikes had the strongest predictive values in the 0–24 hour timeframe with 93% sensitivity, 81% specificity, 87% accuracy and MCC of 0.75 (Table 4).

We also observed that using only one vital sign as an independent predictor of bloodstream infection showed weaker predictive results than using a combination of vital signs. Two or more fever spikes above 39°C had a sensitivity of 52% only, but a specificity of 79%. Two or more HR spikes above two standard deviation age-specific norms had a sensitivity of 59% and a specificity of 60%. One or more MAP dips below 60 mmHg had a sensitivity of 93% and

**Table 4. Predictive analysis using >1 MAP dips and >2 fever spikes or >2 HR spikes.**

| Predictive Metric | Fever Spike Only | HR Spike Only | MAP Dip Only | Combination |
|---|---|---|---|---|
| True Positive Rate (Sensitivity) | 52% | 59% | 93% | 93% |
| True Negative Rate (Specificity) | 79% | 60% | 80% | 81% |
| Positive Predictive Value (Precision) | 72% | 60% | 83% | 83% |
| Negative Predictive Value | 62% | 59% | 92% | 92% |
| False Positive Rate | 21% | 40% | 20% | 19% |
| False Negative Rate | 48% | 41% | 7% | 7% |
| False Discovery Rate | 28% | 40% | 17% | 17% |
| Accuracy | 66% | 60% | 87% | 87% |
| Matthews Correlation Coefficient (MCC) | 0.33 | 0.20 | 0.74 | 0.75 |

**Table 5. Mean fever spikes, HR spikes and MAP dips by infectious organism type.**

| Type of Organism | Blood Culture (N) | Temperature | | | | Heart Rate | | | | Mean Arterial Pressure | | | |
|---|---|---|---|---|---|---|---|---|---|---|---|---|---|
| | | Median (˚C) | Mean Fever Spike (SD) | P-value (G+ vs G-) | P-value (Bact. Vs Fung.) | Median (BPM) | Mean HR Spike (SD) | P-value (G+ vs G-) | P-value (Bact. Vs Fung.) | Median (mmHg) | Mean MAP Dip (SD) | P-value (G+ vs G-) | P-value (Bact. Vs Fung.) |
| **Bacteria** (Bact.) | 106 | 38.2 | 2.82 ± 3.77 | 5E-01 | 5E-01 | 140.8 | 13.0 ± 9.77 | 4E-02[a] | 2E-01 | 71.6 | 4.61 ± 6.45 | 8E-02 | 7E-01 |
| **Gram+** (G+) | 29 | 38.2 | 2.45 ± 2.41 | | | 139.2 | 9.83 ± 5.99 | | | 73.9 | 2.83 ± 4.60 | | |
| **Gram-** (G-) | 77 | 38.2 | 2.96 ± 4.18 | | | 141.4 | 14.2 ± 10.7 | | | 70.7 | 5.29 ± 6.93 | | |
| **Fungi** (Fung.) | 19 | 38.1 | 2.16 ± 2.97 | | | 137.2 | 9.63 ± 6.21 | | | 71.5 | 3.95 ± 7.07 | | |

[a]Statistical significance using Student's t-test

specificity of 80%. Using the combination scoring criteria, we observed high specificity and sensitivity across the 0–24 hour timeframe post-blood culture draw.

## Infectious organisms

The dynamic trends in vital sign patterns and the presence of infectious microorganisms were also evaluated from the infection group (Table 5). A total of 125 infectious microorganisms were identified from the blood cultures with one or more positive bloodstream infections. A majority of the infectious microorganisms were Gram-negative (62%), followed by Gram-positive (23%) and fungi (15%). The most common Gram-negative bacteria species were *A. baumannii* (30%), followed by *K. pneumonia* (17%), and *P. aeruginosa* (17%). There were statistically significant differences in HR spikes between patients with Gram-negative and Gram-positive bacteria. There were no statistically significant differences between the common Gram-negative species in the mean number of fever spikes, HR spikes and MAP dips (Table 6).

## Prediction of gram-negative bacteria using vital sign patterns

The mean number of fever spikes, HR spikes, and MAP dips also appeared to distinguish Gram-negative bacteria from Gram-positive bacteria and fungi, albeit much lower than the >90% accuracy obtained in predicting bloodstream infection. We evaluated each vital sign individually, as well as a combination of all three, to predict the presence of Gram-negative

**Table 6. Mean fever spikes, HR spikes and MAP dips by most common gram-negative bacteria.**

| Gram-Bacteria | Blood Culture (N) | Temperature | | | | Heart Rate | | | | Mean Arterial Pressure | | | |
|---|---|---|---|---|---|---|---|---|---|---|---|---|---|
| | | Median (˚C) | Mean Fever Spike (SD) | P-value (A.bau vs K. pne) | P-value (A.bau vs P.aer) | Median (BPM) | Mean HR Spike (SD) | P-value (A.bau vs K. pne) | P-value (A.bau vs P.aer) | Median (mmHg) | Mean MAP Dip (SD) | P-value (A.bau vs K. pne) | P-value (A.bau vs P.aer) |
| **A. baumannii** (A.bau) | 23 | 38.1 | 2.48 ± 3.68 | 6E-01 | 2E-01 | 138.6 | 15.2 ± 12.2 | 3E-01 | 4E-01 | 70.2 | 6.70 ± 8.17 | 2E-01 | 9E-01 |
| **K. pneumoniae** (K.pne) | 13 | 38.3 | 3.08 ± 3.55 | | | 146.4 | 11.4 ± 7.40 | | | 73.9 | 3.08 ± 5.06 | | |
| **P. aeruginosa** (P.aer) | 13 | 38.2 | 4.77 ± 6.39 | | | 140.5 | 19.3 ± 13.1 | | | 70.1 | 7.00 ± 9.50 | | |

**Table 7. Predictive analysis of Gram-negative bacteria using >5 MAP dips and >14 HR spikes or >2 fever spikes.**

| Predictive Metric | Fever Spike Only | HR Spike Only | MAP Dip Only | Combination |
|---|---|---|---|---|
| True Positive Rate (Sensitivity) | 52% | 36% | 34% | 57% |
| True Negative Rate (Specificity) | 44% | 48% | 21% | 44% |
| Positive Predictive Value (Precision) | 60% | 53% | 41% | 62% |
| Negative Predictive Value | 36% | 32% | 16% | 39% |
| False Positive Rate | 56% | 52% | 79% | 56% |
| False Negative Rate | 48% | 64% | 66% | 43% |
| False Discovery Rate | 40% | 47% | 59% | 38% |
| Accuracy | 49% | 41% | 29% | 52% |
| Matthews Correlation Coefficient (MCC) | -0.04 | -0.15 | -0.44 | 0.01 |

bacteria in positive blood cultures. The combination of 5 or more MAP dips and 14 or more HR spikes or 2 or more fever spikes had the highest predictive values for Gram-negative bacteria with 57% sensitivity, 44% specificity, 52% accuracy and MCC of 0.01 (Table 7).

## Discussion

Our findings revealed that a combination of fever spikes above 39°C, HR spikes above two standard deviation age-specific norms, and MAP dips below 60 mmHg predicted positive bloodstream infection within 0–24 hour timeframe post-blood culture draw. In a previous study, our group found that using two or more fever spikes above 39°C had 89% specificity, 56% sensitivity, 72% accuracy and MCC of 0.47 within the 0–24 hour timeframe post-excisional surgery blood culture draw [21]. However, the predictive metric was limited to temperature data and blood cultures following excisional surgery only, and sensitivity was low. Our current study implemented multiple observed variables including HR and MAP patterns, as well as multiple blood cultures drawn during the patients' stay in the pediatric burn unit. This addition enhanced our simple scoring criteria and improved predictive accuracy with a 93% sensitivity, 81% specificity, 87% accuracy and MCC of 0.75. With these levels, we suggest the metrics are candidates for clinical use in resource-poor environments, or as a guidance for preemptive antibiotic use in some cases.

This study also suggests that spikes in vital sign patterns have potential for utilization in predicting Gram-negative bacteria in positive blood cultures. After testing various combinations and number of spikes, we identified that the combination of 5 or more MAP dips and 14 or more HR spikes or 2 or more fever spikes had the highest predictive values for Gram-negative bacteria with 57% sensitivity, 44% specificity, 52% accuracy and MCC of 0.01. This was the strongest sensitivity and specificity found across all combination criteria tested. We believe that the simple scoring system has a lower specificity than sensitivity because a few of the patients tend to narrowly miss the cutoff for spikes. It is possible that a longer observation period may increase the accuracy of the cutoff for spikes, which has been the case for predicting infections in our prior work [21]. It may also prove useful to explore temporal trends beyond a simple spike count to boost accuracy.

Patients in the infection group had significantly greater TBSA burns, full thickness burns, LOS, ventilation days and ICU days compared to the control group. With larger-size TBSA burns and full-thickness burns, there is a greater likelihood of developing bloodstream infections [10, 11]. LOS, ventilation days and ICU days are typically extended in the cases of bloodstream infections for additional monitoring according to our institution's discharge protocol.

A majority of patients in this study were exposed to antipyretic medications. The standard care of treatment at our institution is administering antipyretic medications for the purpose of reducing fever and not directly for pain control. There is a possibility that the use of antipyretic medications may have affected body temperature. However, our results suggests that even in the presence of antipyretic medications, vital sign fluctuations may be utilized by clinical staff to identify bloodstream infections.

At our institution, patients were started on empiric antibiotics if they presented high fevers and low MAPs until the results from blood cultures were confirmed to be positive or negative. After completing the empiric antibiotics course for positive bloodstream infections, our standard care of treatment involved a routine negative screening culture practice until patients were discharged from the pediatric burn unit. Even with the use of empiric antibiotics, our prediction method detected abnormal vital sign patterns, as our combination scoring criteria had low false negative (7%) and false positive rates (19%). This suggests that the use of antibiotics does not significantly affect vital sign fluctuations nor the accuracy of the predictive test metric.

In the infection group, 38 patients were confirmed to be on antibiotics during the study timeframe. We were unable to confirm the use of antibiotics in 11 patients in the infection group due to limited access to medical records prior to 2011. In the control group, 35 patients were confirmed to be on antibiotics during the study timeframe, and 11 patients were confirmed to not require antibiotics during their inpatient stay.

Previous sepsis scoring systems have established specific parameters to predict the likelihood of developing bloodstream infections. The current Sepsis-3 criteria clinically operationalizes sepsis as an increase in the Sequential Organ Failure Assessment (SOFA) score by +2 points [22]. The SOFA criteria assigns +1 points to a patient with any of the following: an altered mental status, respiratory rate over 22 breaths per minute, or a systolic blood pressure under 100 mmHg [22–24]. Similarly to the SOFA criteria, the SIRS criteria assigns +1 points to a patient with any of the following: temperature outside of the 36 to 38˚C range, heart rate over 90 BPM, respiratory rate over 20 breaths per minute, and white blood count outside of the 4000 to 12,000/mm³ range [25]. For the Sepsis-3, SOFA and SIRS criteria, the presence of two or more parameters indicates likelihood of developing sepsis.

To address the concern for burn-specific criteria to accurately predict bloodstream infections, The American Burn Association (ABA) proposes that the presence of three or more of the following parameters indicate sepsis: temperature of >39˚C or <36.5˚C, progressive tachycardia, progressive tachypnea, thrombocytopenia, hyperglycemia (without preexisting diabetes mellitus), and inability to continue enteral feedings over 24 hours [26]. After presenting any three parameters, clinical staff should initiate empiric antibiotic administration and follow up with blood culture tests to document bloodstream infection [27]. Following the ABA and SIRS criteria, one study conducted by Mann-Salinas established novel clinical predictors of bloodstream infection in burn patients, including heart rate >130 beats per min, mean arterial pressure <60 mm Hg, base deficit <−6 mEq/L, temperature <36˚C, use of vasoactive medications, and glucose >150 mg/dl [28].

However, these sepsis scoring criteria (ABA, Sepsis-3 and Mann-Salinas) have rarely been used in clinic settings because their predictive accuracy have been found to be lower in the literature [29]. In one study that validated these sepsis scoring criteria, the ABA and Mann-Salinas criteria had an overall predictive accuracy of 59% and 28%, respectively, compared to prospective clinical diagnoses by burn-specialist physicians [29]. The Sepsis-3 criteria had the greatest overall predictive accuracy of the three existing criteria (85%; $p<0.05$). However, it is limited to utilization in the general ICU setting and harder to evaluate as an early indicator of sepsis in patients with more severe burn injuries [29].

Other studies that use specific clinical biomarkers to predict sepsis involve additional laboratory tests and equipment to improve diagnostic accuracy and specificity. In a meta-analysis evaluating the use of procalcitonin and C-reactive protein in the sepsis diagnosis of adult patients, procalcitonin had an 85% specificity and 77% sensitivity, and C-reactive protein had an 80% specificity and 61% sensitivity [30]. Compared to these biomarkers limited in sensitivity and specificity, our predictive metric uses readily available vital signs and requires only simple counting, which is ideal in resource-poor environments and provides practical supporting information if such laboratory tests are not available on time. These characteristics of the test metric allow feasible implementation in the clinic setting without requiring excessive levels of laboratory data, as well as providing high specificity and sensitivity for diagnosing bloodstream infection in pediatric burn patients.

## Limitations

The generalizability of these findings are limited by the nature of the study design, as it is a retrospective review of a single institution's experience. The variables (fever spikes, HR spikes and MAP dips) all occurred within same 24 hour study timeframes, however the variables did not occur within an hour of each other. While we did not detect any immediate trends between the timing of measurements and sepsis accuracy, future studies would benefit from a larger sample size. The exact timing of the measurements depends on a large number of factors and notions in the clinic setting, and therefore it would take a significantly larger patient study sample to be able to analyze such trends. We are currently working on acquiring multi-hospital data sets to achieve such analyses. The study was also limited due to incomplete data entries, which did not allow our group to analyze other vital signs such as respiratory rate, fraction of inspired oxygen, or basic lab analysis such as white blood cell count. Biomarker data, such as procalcitonin, C-reactive protein, tumor necrosis factor-α, interleukin-6, and interleukin-10, was also not available, which could be explored in future research to enhance accuracy or allow detection within a period less than 24 hours. Timeframes less than 24 hours were not evaluated for the predictive metric due to insufficient data entries recorded within timeframes less than 24 hours. Future studies would benefit from evaluating a shorter study timeframe where vital signs are collected more frequently such that assessments can be done within hours. Our group is currently developing a larger follow-up prospective clinical study with such characteristics.

## Conclusion

This study found that using a combination scoring criteria of fever spikes, HR spikes and MAP dips predicted bloodstream infection in pediatric patients with burn injuries with 87% accuracy, which may justify its use in resource-poor environments, or in cases where practical supporting evidence is needed for preemptive antibiotic treatment before culture results are available. Moreover, our approach shows potential to be utilized in predicting the presence of Gram-negative bacteria in positive blood cultures.

## Author Contributions

**Conceptualization:** Robert Leo Sheridan, Korkut Uygun.

**Data curation:** Farzin Sadeq, Jonah Poster, Chris Chu.

**Formal analysis:** Farzin Sadeq, Jonah Poster, Chris Chu.

**Funding acquisition:** Robert Leo Sheridan, Korkut Uygun.

**Investigation:** Farzin Sadeq, Jonah Poster, Chris Chu.

**Methodology:** Farzin Sadeq, Jonah Poster, Chris Chu.

**Project administration:** Farzin Sadeq.

**Resources:** Robert Leo Sheridan, Korkut Uygun.

**Supervision:** Robert Leo Sheridan, Korkut Uygun.

**Validation:** Robert Leo Sheridan, Korkut Uygun.

**Visualization:** Farzin Sadeq.

**Writing – original draft:** Farzin Sadeq.

**Writing – review & editing:** Farzin Sadeq, Jonah Poster, Joan Weber, Martha Lydon, Maggie Dylewski Begis, Robert Leo Sheridan, Korkut Uygun.

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
