## [Decision Letter · Decision Letter 0]

18 Oct 2021

PONE-D-21-31157Specific patterns of vital sign fluctuations predict bloodstream infection in pediatric burn patientsPLOS ONE

Dear Dr. Uygun,

Thank you for submitting your manuscript to PLOS ONE. After careful consideration, we feel that it has merit but does not fully meet PLOS ONE’s publication criteria as it currently stands. Therefore, we invite you to submit a revised version of the manuscript that addresses the points raised during the review process.

We look forward to receiving your revised manuscript.

Kind regards,

David G. Greenhalgh, MD

Academic Editor

PLOS ONE

Journal Requirements:

Additional Editor Comments:

Academic Editor: Please address the concerns of the reviewers and we will re-evaluate the paper.

Reviewers' comments:

Reviewer's Responses to Questions

**Comments to the Author**

1. Is the manuscript technically sound, and do the data support the conclusions?

Reviewer #1: Yes

Reviewer #2: Yes

2. Has the statistical analysis been performed appropriately and rigorously? 

Reviewer #1: Yes

Reviewer #2: Yes

3. Have the authors made all data underlying the findings in their manuscript fully available?

Reviewer #1: Yes

Reviewer #2: Yes

4. Is the manuscript presented in an intelligible fashion and written in standard English?

Reviewer #1: Yes

Reviewer #2: Yes

5. Review Comments to the Author

Reviewer #1: This is an interesting article that highlights the importance and difficulty of diagnosing sepsis in the setting of severe burn injury. I have some concerns for the authors:

1. It seems from the data that MAP dips are the most relevant correlation with sepsis. I suggest that a multivariate regression analysis would provide a better insight into the weight or importance of MAP dips relative to other variables.

2. What is the time frame between MAP dips and other variables (fever, HR spikes)? In other words, do these events occur within a similar time frame or do they only occur within the same 24 hour period but do not occur within an hour of each event.

3. Does the time frame between each variable event correlate to sepsis prediction or accuracy? For example, do events that occur relatively close together (fever, HR spikes, and MAP dips) lead to a more accurate sepsis prediction?

Reviewer #2: The authors presented a manuscript examining the fluctuation in patterns of simple vital signs data to predict “diagnose” blood stream infection in paediatric burn patients. They included retrospective data from 200 children with >20% TBSA. This work builds on their previously published research of the role of temperature patterns to predict blood stream infection.

I thoroughly enjoyed reading the manuscript, it is well written, referenced, and a good study design despite the package of limitations that comes with retrospective studies. The strength of the study is taking positive blood culture as the “golden standard/reference standard” for defining sepsis rather than other scoring systems that are either not validated or have low reliability. Others have used clinical diagnosis of burn specialist surgeons and critical care physicians as the reference standards for validation to avoid this problem. Furthermore, all these current sepsis scores are difficult to apply to paediatric burns patients.

Early diagnosis of sepsis and immediately initiating management was shown to save many lives of our patients. In future, advancement of bedside microbiological sequencing may give all clues that clinicians need for our sick burn patient. However, until then, a simple clinical algorithm would be extremely useful to diagnose sepsis, confidently and reliably. This manuscript is an important step in this path and may lead to a simple global sepsis scoring criteria for early paediatric burn sepsis diagnosis.

I’d congratulate the author on perusing a simple solution for a very difficult problem and I am looking forward seeing the data from their prospective ongoing project.

My comments are:

1. Title: I fully understand the authors are factual and careful with the title wording. As I personally believe this study would be a step forward in sepsis diagnosis in paediatric burn patients, I would suggest using the keywords (sepsis, diagnosis), to be straight to the point.

2. The whole dilemma of burn sepsis scorings is ‘sepsis definition’ and setting ‘Reference Standard/Golden Standard’ to check the reliability of a new scoring system. The authors used Positive Blood Culture (PBC), which is superior to other unvalidated or unreliable scores as definition, however, there are problems regarding interpretations of PBC in clinical practice.

Would the author please elaborate why PBC was used and its limitations?

Would the authors discuss their views using the “clinical diagnosis of sepsis by burn specialist surgeon and critical care physician” for sepsis definition as reported by Yan et al, 2018, as a reference standard for future trials?

3. The authors have explained the use of empiric antibiotics after taking blood cultures and the effect on vital signs. Would they please elaborate on how many started the therapy and when, in each group?

4. It would be interesting to the reader if the authors elaborate on why this simple scoring system has lower specificity than sensitivity? Would that be related to the use of PBC as the ‘Reference Standard’?

6. PLOS authors have the option to publish the peer review history of their article (what does this mean?). If published, this will include your full peer review and any attached files.

Reviewer #1: No

Reviewer #2: **Yes: **Naiem S Moiemen

---

## [Author Response · Author response to Decision Letter 0]

13 Dec 2021

Reviewer #1: This is an interesting article that highlights the importance and difficulty of diagnosing sepsis in the setting of severe burn injury. I have some concerns for the authors:

Thank you for your insightful comments to add clarity on the study methods and analyses for our article. Please see our comments below. 

1. It seems from the data that MAP dips are the most relevant correlation with sepsis. I suggest that a multivariate regression analysis would provide a better insight into the weight or importance of MAP dips relative to other variables.

Author Response: We have performed two multivariate regression analyses to determine the weight of each independent variable (fever spikes, HR spikes and MAP dips) relative to the other variables. Our regression analyses found that HR spikes were significant and held more statistical weight in the prediction metric than MAP dips or fever spikes. 

Location of Revision: 124-125, 207-211

2. What is the time frame between MAP dips and other variables (fever, HR spikes)? In other words, do these events occur within a similar time frame or do they only occur within the same 24 hour period but do not occur within an hour of each event.

Author Response: The variables (fever spikes, HR spikes and MAP dips) all occurred within the same 24 hour study timeframes, however the variables did not occur within an hour of each other.

Location of Revision: 353-355

3. Does the time frame between each variable event correlate to sepsis prediction or accuracy? For example, do events that occur relatively close together (fever, HR spikes, and MAP dips) lead to a more accurate sepsis prediction?

Author Response: While we did not detect any immediate trends between the timing of measurements and sepsis accuracy, future studies would benefit from a larger sample size. The exact timing of the measurements depends on a large number of factors and notions in the clinic setting, and therefore it would take a significantly larger patient study sample to be able to analyze such trends. We are currently working on acquiring multi-hospital data sets to achieve such analyses.

Location of Revision: 355-360

Reviewer #2: The authors presented a manuscript examining the fluctuation in patterns of simple vital signs data to predict “diagnose” blood stream infection in paediatric burn patients. They included retrospective data from 200 children with >20% TBSA. This work builds on their previously published research of the role of temperature patterns to predict blood stream infection. 

I thoroughly enjoyed reading the manuscript, it is well written, referenced, and a good study design despite the package of limitations that comes with retrospective studies. The strength of the study is taking positive blood culture as the “golden standard/reference standard” for defining sepsis rather than other scoring systems that are either not validated or have low reliability. Others have used clinical diagnosis of burn specialist surgeons and critical care physicians as the reference standards for validation to avoid this problem. Furthermore, all these current sepsis scores are difficult to apply to paediatric burns patients. Early diagnosis of sepsis and immediately initiating management was shown to save many lives of our patients. In future, advancement of bedside microbiological sequencing may give all clues that clinicians need for our sick burn patient. However, until then, a simple clinical algorithm would be extremely useful to diagnose sepsis, confidently and reliably. This manuscript is an important step in this path and may lead to a simple global sepsis scoring criteria for early paediatric burn sepsis diagnosis. 

I’d congratulate the author on perusing a simple solution for a very difficult problem and I am looking forward seeing the data from their prospective ongoing project. My comments are:

Thank you for your insightful comments to add clarity on the study methods and discussion for our article. Please see our comments below. 

1. Title: I fully understand the authors are factual and careful with the title wording. As I personally believe this study would be a step forward in sepsis diagnosis in paediatric burn patients, I would suggest using the keywords (sepsis, diagnosis), to be straight to the point.

Author Response: Thank you for bringing this to our attention. We have amended the title of the article to “Specific patterns of vital sign fluctuations predict infection and enable sepsis diagnosis in pediatric burn patients” to further clarify the main objectives of the study. 

Location of Revision: 4-5

2. The whole dilemma of burn sepsis scorings is ‘sepsis definition’ and setting ‘Reference Standard/Golden Standard’ to check the reliability of a new scoring system. The authors used Positive Blood Culture (PBC), which is superior to other unvalidated or unreliable scores as definition, however, there are problems regarding interpretations of PBC in clinical practice. Would the author please elaborate why PBC was used and its limitations? Would the authors discuss their views using the “clinical diagnosis of sepsis by burn specialist surgeon and critical care physician” for sepsis definition as reported by Yan et al, 2018, as a reference standard for future trials? 

Author Response: Positive blood cultures are used at our pediatric burn institution as the current reference standard because they are superior in sepsis accuracy, compared to other unvalidated scoring systems. However, it is also possible to miss infections using blood cultures alone, and the clinical team will often order pan (sputum and urine) cultures for further accuracy.

Before obtaining the positive blood cultures, the clinical team correctly administered empiric antibiotics for 38 patients in the infection group. We were unable to confirm antibiotic use in 11 patients due to limited access to medical records prior to 2011. The clinical team incorrectly administered empiric antibiotics for 35 patients in the control group, which were later confirmed by the routine negative blood cultures. The clinical team correctly diagnosed 11 patients that did not require empiric antibiotics during their inpatient stay. 

The Yan et al. study suggests that using clinical diagnoses made by the clinical team as the reference standard is a strong approach for timely identification of sepsis in burn patients. We agree with a similar approach reported by Yan et al., as clinician diagnoses are based on clinical signs of sepsis, such as vital sign fluctuations. The main motivation behind this current study was to develop a rigorous predictive metric that statistically interpreted these clinical signs of sepsis already used by the clinical team to enable sepsis diagnoses.

3. The authors have explained the use of empiric antibiotics after taking blood cultures and the effect on vital signs. Would they please elaborate on how many started the therapy and when, in each group?

Author Response: In the infection group, 38 patients were confirmed to be on antibiotics during the study timeframe. We were unable to confirm the use of antibiotics in 11 patients in the infection group due to limited access to medical records prior to 2011. In the control group, 35 patients were confirmed to be on antibiotics during the study timeframe, and 11 patients were confirmed to not require antibiotics during their inpatient stay.

Location of Revision: 301-305

4. It would be interesting to the reader if the authors elaborate on why this simple scoring system has lower specificity than sensitivity? Would that be related to the use of PBC as the ‘Reference Standard’?

Author Response: We believe that the simple scoring system has a lower specificity than sensitivity because a few of the patients tend to narrowly miss the cutoff for spikes. It is possible that a longer observation period may increase the accuracy of the cutoff for spikes, which has been the case for predicting infections in our prior work. 

Location of Revision: 272-275

---

## [Decision Letter · Decision Letter 1]

19 Jan 2022

Specific patterns of vital sign fluctuations predict infection and enable sepsis diagnosis in pediatric burn patients

PONE-D-21-31157R1

Dear Dr. Uygun,

We’re pleased to inform you that your manuscript has been judged scientifically suitable for publication and will be formally accepted for publication once it meets all outstanding technical requirements.

Kind regards,

David G. Greenhalgh, MD

Academic Editor

PLOS ONE

Additional Editor Comments (optional):

Reviewers' comments:

Reviewer's Responses to Questions

**Comments to the Author**

1. If the authors have adequately addressed your comments raised in a previous round of review and you feel that this manuscript is now acceptable for publication, you may indicate that here to bypass the “Comments to the Author” section, enter your conflict of interest statement in the “Confidential to Editor” section, and submit your "Accept" recommendation.

Reviewer #1: All comments have been addressed

Reviewer #2: All comments have been addressed

2. Is the manuscript technically sound, and do the data support the conclusions?

Reviewer #1: Yes

Reviewer #2: Yes

3. Has the statistical analysis been performed appropriately and rigorously? 

Reviewer #1: Yes

Reviewer #2: Yes

4. Have the authors made all data underlying the findings in their manuscript fully available?

Reviewer #1: Yes

Reviewer #2: Yes

5. Is the manuscript presented in an intelligible fashion and written in standard English?

Reviewer #1: Yes

Reviewer #2: Yes

6. Review Comments to the Author

Reviewer #1: Thank you for addressing all of my concerns. The paper is well written and worthy of publication. This adds to the current literature on identifying sepsis in burn patients.

Reviewer #2: Thanks for addressing all the comments. Surely this has strengthened the manuscript. I believe this work would be a good addition to the literature that is definitely in need of similar contributions.

I hope your finding would be validated in a multi-centre trial in the near future.

Congratulations for the good work

7. PLOS authors have the option to publish the peer review history of their article (what does this mean?). If published, this will include your full peer review and any attached files.

Reviewer #1: No

Reviewer #2: **Yes: **Dr. Naiem Moiemen

---

## [Editor Report · Acceptance letter]

28 Jan 2022

PONE-D-21-31157R1 

Specific patterns of vital sign fluctuations predict infection and enable sepsis diagnosis in pediatric burn patients 

Dear Dr. Uygun:

I'm pleased to inform you that your manuscript has been deemed suitable for publication in PLOS ONE. Congratulations! Your manuscript is now with our production department. 

Kind regards, 

on behalf of

Dr. David G. Greenhalgh 

Academic Editor

PLOS ONE